# Continuous Visual Autoregressive Generation via Score Maximization

Chenze Shao [1]   Fandong Meng [1]   Jie Zhou [1]

## Abstract

Conventional wisdom suggests that autoregressive models are used to process discrete data. When applied to continuous modalities such as visual data, Visual AutoRegressive modeling (VAR) typically resorts to quantization-based approaches to cast the data into a discrete space, which can introduce significant information loss. To tackle this issue, we introduce a Continuous VAR framework that enables direct visual autoregressive generation without vector quantization. The underlying theoretical foundation is strictly proper scoring rules, which provide powerful statistical tools capable of evaluating how well a generative model approximates the true distribution. Within this framework, all we need is to select a strictly proper score and set it as the training objective to optimize. We primarily explore a class of training objectives based on the energy score, which is likelihood-free and thus overcomes the difficulty of making probabilistic predictions in the continuous space. Previous efforts on continuous autoregressive generation, such as GIVT and diffusion loss, can also be derived from our framework using other strictly proper scores. Source code: https://github.com/shaochenze/EAR.

## 1. Introduction

Autoregressive large language models (Achiam et al., 2023; Touvron et al., 2023; Team et al., 2023; Bai et al., 2023) have demonstrated remarkable scalability and generalizability in understanding and generating discrete text, which has inspired the exploration of autoregressive generation on other data modalities. However, autoregressive models equipped with cross-entropy loss are limited to handle discrete tokens from a finite vocabulary. Therefore, for continuous modalities such as visual data, Visual AutoRegressive modeling (VAR[1]) typically resort to quantization-based approaches (van den Oord et al., 2017; Razavi et al., 2019; Esser et al., 2021; Yu et al., 2024a) to cast the data into a discrete space.

Discrete visual representation based on vector quantization provides support for autoregressive generation, yet the primary concern lies in the information loss due to quantization errors. During visual generation, quantization errors degrades the reconstruction quality of discrete image tokenizers, which upper-bounds the generation quality (Rombach et al., 2022). Moreover, discrete representations compromise the model's perception of low-level details, restricting its ability to capture continuous variations and subtle differences. Consequently, in terms of visual understanding, the performance of discrete tokenizers often lags behind that of continuous tokenizers (Wu et al., 2024; Xie et al., 2024).

Given the limitations associated with vector quantization, there is a growing interest in continuous visual autoregressive generation. However, without a finite vocabulary, it is generally intractable to explicitly predict the likelihood over continuous spaces and train with likelihood maximization. Prior to this work, autoregressive generation in continuous spaces has been explored through GIVT (Tschannen et al., 2023) and diffusion loss (Li et al., 2024). Nevertheless, the expressive capability of GIVT is confined to the pre-defined family of Gaussian mixtures (Tschannen et al., 2023), and the per-token diffusion procedure necessitates multiple denoising iterations to recover the token distribution, which significantly increases the inference latency (Li et al., 2024).

In this work, we introduce a Continuous VAR framework that enables direct visual autoregressive generation without vector quantization. The underlying theoretical foundation is strictly proper scoring rules (Brier, 1950; Good, 1952; Gneiting & Raftery, 2007), which provide powerful statistical tools capable of evaluating how well a generative model approximates the true distribution. Specifically, scoring rules are functions to assess the quality of a probability distribution based on the observed sample. A scoring rule is considered strictly proper if it encourages the model to make honest predictions. In other words, the expected score

---

[1]Pattern Recognition Center, WeChat AI, Tencent Inc. Correspondence to: Chenze Shao <chenzeshao@tencent.com>, Fandong Meng <fandongmeng@tencent.com>, Jie Zhou <withtomzhou@tencent.com>.

*Proceedings of the 42$^{nd}$ International Conference on Machine Learning*, Vancouver, Canada. PMLR 267, 2025. Copyright 2025 by the author(s).

---

[1]Here, VAR refers to the autoregressive modeling of visual content, not limited to next scale prediction (Tian et al., 2024).

is maximized only when model predictions follow the true distribution, and any deviation from the truth will result in a decrease in expected score.

The intrinsic property of strictly proper scoring rules makes them well-suited training objectives for generative models. A prominent example is the cross-entropy loss used in discrete autoregressive models, which corresponds to the maximization of logarithmic score (Good, 1952). Within the Continuous VAR framework, all we need is to select a strictly proper score for continuous variables and set it as the training objective to optimize. Previous efforts on continuous autoregressive generation, such as GIVT and diffusion loss, can also be derived from this framework, where they are respectively aligned with the logarithmic score (Good, 1952) and the Hyvärinen score (Hyvärinen, 2005).

Under the Continuous VAR framework, we primarily explore a class of training objectives based on the energy score (Székely, 2003), which is likelihood-free and thus overcomes the difficulty of making probabilistic predictions in the continuous space. The associated energy loss incentivizes the model to generate samples close to the target label, while maintaining the diversity between independent samples. The model architecture remains largely analogous to a discrete Transformer (Vaswani et al., 2017), with the key difference being the substitution of the softmax layer with a small MLP generator, both of which transform the hidden representation into a distribution. Similar to Generative Adversarial Networks (Goodfellow et al., 2014), the MLP generator is an implicit generative model that takes random noises as additional inputs, and the predictive distribution is implicitly represented by its sampling process.

Experiments on the ImageNet 256×256 benchmark (Deng et al., 2009) show that our approach achieves stronger visual generation quality than the traditional autoregressive Transformer that uses a discrete tokenizer. Compared to diffusion-based methods, our approach exhibits substantially higher inference efficiency, as it does not require multiple denoising iterations to recover the target distribution.

## 2. Related Work

**Visual Autoregressive Generation**. Early efforts approached visual autoregressive generation by treating the image as a sequence of pixels (Gregor et al., 2014; Parmar et al., 2018; van den Oord et al., 2016a;b). To mitigate the expensive cost of autoregressive modeling at the pixel level, van den Oord et al. (2017) introduced the vector quantization technique to represent an image as a set of discrete tokens, paving the way for more effective autoregressive image generation with both causal (Razavi et al., 2019; Esser et al., 2021; Ramesh et al., 2021; Yu et al., 2022; Sun et al., 2024a; Tian et al., 2024) and masked Transformers

(Chang et al., 2022; Li et al., 2023; Chang et al., 2023). However, the information loss incurred during the quantization process becomes a bottleneck for the generation quality. Consequently, recent focus has shifted towards finding a better image tokenizer (Yu et al., 2022; Mentzer et al., 2024; Yu et al., 2024a;b; Weber et al., 2024). In parallel, there is a growing interest in employing continuous tokenizers for autoregressive image generation, with Gaussian mixture models (Tschannen et al., 2023) and diffusion models (Li et al., 2024) being used to represent the token distribution. Our approach advances this direction by establishing a universal framework for predicting continuous tokens.

**Strictly Proper Scoring Rules**. Strictly proper scoring rules, initially introduced in Brier (1950) for the verification of weather forecasts, have since evolved into a comprehensive theoretical framework for evaluating probabilistic forecasts. In the realm of deep learning, the most extensively applied scoring rule is the logarithmic score (Good, 1952), which is closely linked with maximum likelihood estimation, cross-entropy loss, and perplexity evaluation. The Brier score is also widely used for training classification networks (Shoemaker, 1991; Hung et al., 1996; Kline & Berardi, 2005; Hui & Belkin, 2021) and evaluating their calibration (Lakshminarayanan et al., 2017; Ovadia et al., 2019; Gruber & Buettner, 2022). Recently, Shao et al. (2024) proposed using scoring rules as the training objective for autoregressive language modeling. In the continuous space, the Hyvärinen score (Hyvärinen, 2005) plays an important role in score matching, which gives rise to score-based diffusion models (Song & Ermon, 2019; Song et al., 2021). The energy score has also been employed in generative modeling, with applications spanning image generation (Bellemare et al., 2018), speech synthesis (Gritsenko et al., 2020), time series prediction (Pacchiardi et al., 2024; Pacchiardi & Dutta, 2022), and self-supervised learning (Vahidi et al., 2024). In this work, we focus on leveraging scoring rules to enable the autoregressive modeling of continuous data.

## 3. Continuous Visual Autoregressive Generation

In this section, we begin by introducing the essential background of strictly proper scoring rules. Following that, we present the Continuous VAR framework, which enables continuous visual autoregressive generation via score maximization. For simplicity of notation, we assume a setting of unconditional generation, and the conclusion can be extended to conditional generation scenarios.

### 3.1. Strictly Proper Scoring Rules

In statistical decision theory, scoring rules serve as quantitative measures to assess the quality of probabilistic predic-

tions, by assigning a numerical score based on the predicted distribution $p$ and the observed sample $x$. Let $\mathcal{X}$ represents the sample space and $\mathcal{P}$ be the set of probability measures on $\mathcal{X}$. A scoring rule $S$ takes values in the extended real line $\overline{\mathbb{R}} = [-\infty, \infty]$, indicating the reward or utility of predicting $p$ when sample $x$ is observed:

$$S(p, x) : \mathcal{P} \times \mathcal{X} \mapsto \overline{\mathbb{R}}. \tag{1}$$

The role of scoring rules is to assess whether the prediction $p$ honestly represents the underlying sample distribution $q$. This is reflected in the expected score with respect to $x \sim q$, denoted as $S(p, q)$:

$$S(p, q) = \mathbb{E}_{x \sim q}[S(p, x)]. \tag{2}$$

A proper scoring rule should encourage the model to make honest predictions. Formally, a scoring rule is proper if the expected score is maximized when the model reports true probabilities:

$$S(p, q) \leq S(q, q), \quad \forall p, q \in \mathcal{P}. \tag{3}$$

It is strictly proper when the equality holds if and only if $p = q$. The strict propriety means that the score maximizer is unique, where any deviation from the truth will result in a decrease in expected score.

The study of scoring rules, which dates back to the Brier score (Brier, 1950) for the verification of weather forecasts, has evolved into a comprehensive framework offering a wealth of useful scores, such as the logarithmic score (Good, 1952), Brier score (Brier, 1950), and spherical score (Roby, 1965). For continuous variables, the choices expand to strictly proper scores such as the energy score (Székely, 2003), CRPS (Matheson & Winkler, 1976), and Hyvärinen score (Hyvärinen, 2005), as well as proper scores like the kernel score (Eaton, 1981) and Variogram score (Scheuerer & Hamill, 2015). Gneiting & Raftery (2007) provides a comprehensive literature review on scoring rules.

### 3.2. Continuous VAR via Score Maximization

The property of strictly proper scoring rules naturally align with the objective of generative models. With a loss function that promotes the maximization of a strictly proper score, the model will be trained to approximate the data distribution. A direct approach is to take the negative of a strictly proper score as the loss:

$$\mathcal{L}_S(p, x) = -S(p, x). \tag{4}$$

For example, maximizing the logarithmic score $S(p, x) = \log p(x)$ recovers the cross-entropy loss. The emphasis on the strict propriety of the scoring rule is crucial. Unlike strictly proper scores which guarantee a unique optimizer, proper but not strictly proper scores results in a loss function

with multiple potential minimizers, making it challenging for the model to converge to the correct distribution. A trivial but telling example is the constant-valued score. While being technically proper, it does not offer meaningful guidance for model training.

When dealing with intricate samples like long texts, videos, or high-resolution images, direct generation poses significant challenges, which necessitates breaking down the process into several steps for autoregressive modeling. In this case, the direct calculation of Equation 4 is not always feasible, but we can evaluate scoring rules at each time step to calculate the following sequence loss (Shao et al., 2024):

$$\mathcal{L}_S(p, x) = -\sum_{t=1}^{T} S(p(\cdot|x_{<t}), x_t). \tag{5}$$

The expected loss is minimized only when every expected score $S(p(\cdot|x_{<t}), q(\cdot|x_{<t}))$ is maximized, which consequently encourages honest sequential predictions $p = q$. However, for continuous-valued generative models, the explicit likelihood estimation is sometimes intractable due to the lack of a finite vocabulary, which makes the score calculation also infeasible. Under these circumstances, we can adopt an unbiased estimator of Equation 5 as the loss function. Since the expectation of loss remains unchanged, the model will still be trained towards approximating the true distribution.

### 3.3. Examples

Here, we revisit the previous methodologies of continuous visual autoregressive generation, namely GIVT and diffusion loss. We will show that these methods can be derived from the perspective of score maximization, falling within our Continuous VAR framework.

**Example 1** (GIVT, Tschannen et al., 2023). Generative Infinite-Vocabulary Transformers (GIVT) is perhaps the first visual Transformer that directly generates vector sequences with real-valued entries. The loss function for GIVT is still the cross-entropy, which corresponds to the maximization of the logarithmic score $S(p, x) = \log p(x)$. To estimate the likelihood, GIVT employs an invertible flow model (Dinh et al., 2015) to simplify the latent distribution, and then approximates it with a Gaussian Mixture Model (GMM). Recently, this methodology has been adapted for speech synthesis (Lin & HE, 2025). The primary limitation of GIVT lies in its expressive capability, which is constrained by the capacity of GMM and its assumption of channel-wise independence.

**Example 2** (Diffusion Loss, Li et al., 2024). Recently, diffusion loss is proposed to model the per-token distribution by a diffusion procedure, which has soon gained widespread

applications such as video generation (Deng et al., 2024; Liu et al., 2024), text-to-image generation (Fan et al., 2024; Yu et al., 2025), speech synthesis (Turetzky et al., 2024), and multi-modal generation (Sun et al., 2024b). Diffusion models (Sohl-Dickstein et al., 2015; Ho et al., 2020), also known as score-based generative models (Song & Ermon, 2019; Song et al., 2021), require multiple denoising iterations to recover the target distribution, which results in a significant inference latency for per-token diffusion procedures.

From the perspective of score matching[2], one needs to estimate the gradients of the data distribution to reverse a diffusion process with Langevin dynamics, which is facilitated by maximizing the Hyvärinen score (Hyvärinen, 2005):

$$S(p, x) = -(2 \operatorname{tr}(\nabla_x^2 \log p(x)) + |\nabla_x \log p(x)|^2), \quad (6)$$

where $tr(\cdot)$ denotes the trace of a matrix. The expected score is equivalent to the score matching objective up to a constant, which shows that the score is strictly proper:

$$S(p, q) = -\mathbb{E}_{x \sim q}[|\nabla_x \log p(x) - \nabla_x \log q(x)|^2] + C(q). \quad (7)$$

The diffusion training objective is an estimation of $S(p, q)$ through denoising score matching over multiple noise scales (Vincent, 2011; Song & Ermon, 2019). It implies that the per-token diffusion loss also falls within our Continuous VAR framework with respect to the Hyvärinen score.

# 4. Energy-based Autoregressive Generation

Under the Continuous VAR framework, we develop a Energy-based AutoRegression (EAR) approach via maximizing the energy score (Székely, 2003). The estimation of energy score does not require explicit likelihood estimations but merely the capability to sample from the model distribution. This reduction of constraints facilitates the design of a more expressive energy Transformer. Moreover, the energy Transformer is highly efficient in inference, capable of predicting the next continuous token in a single forward pass. Further details are elaborated below.

## 4.1. Energy Loss

The energy score is a family of strictly proper scoring rules for continuous variables in $\mathbb{R}^d$. To avoid symbol confusion, we denote the samples drawn from the model distribution $p$ as $x$, and the samples from the data distribution $q$ as $y$. Let $\alpha \in (0, 2)$, the energy score is defined as:

$$S(p, y) = \mathbb{E}[|x_1 - x_2|^\alpha] - 2\mathbb{E}[|x - y|^\alpha], \quad (8)$$

where $x_1, x_2, x \in \mathbb{R}^d$ are independent samples with distribution $p$. The expected energy score is associated with the generalized energy distance $\mathcal{E}^\alpha(p, q)$ by a constant:

$$\begin{aligned} \mathcal{E}^\alpha(p, q) &= 2\mathbb{E}[|x - y|^\alpha] - \mathbb{E}[|x_1 - x_2|^\alpha] - \mathbb{E}[|y_1 - y_2|^\alpha] \\ &= -S(p, q) + C(q). \end{aligned} \quad (9)$$

For $\alpha \in (0, 2)$, $\mathcal{E}^\alpha(p, q) \geq 0$ with equality to zero if and only if $p = q$ (Székely, 2003; Székely & Rizzo, 2013), which implies that the energy score is strictly proper. Note that the distance $\mathcal{E}^2(p, q) = |\mathbb{E}[x] - \mathbb{E}[y]|^2$ is minimized as long as their expectations match, so the energy score at $\alpha = 2$ is proper but not strictly proper.

The energy score can be unbiasedly estimated using two independent samples $x_1, x_2$ drawn from the distribution $p$. In this way, the energy loss is defined as:

$$\mathcal{L}(p, y) = |x_1 - y|^\alpha + |x_2 - y|^\alpha - |x_1 - x_2|^\alpha, \quad (10)$$

which incentivizes the model to generate samples close to the target label, while maintaining the diversity between independent samples. Notably, the energy loss does not require explicit likelihood estimations but merely the capability to sample from the model distribution, which gives much flexibility in the following architecture design of energy Transformer.

## 4.2. Energy Transformer

Figure 1 illustrates the architecture of energy Transformer, where the energy loss is employed to supervise each autoregressive generation step. The continuous-valued energy Transformer remains largely analogous to a discrete Transformer (Vaswani et al., 2017), with the key difference being the substitution of the softmax layer with a small MLP generator, both of which transform the hidden representation into a distribution. Similar to Generative Adversarial Networks (Goodfellow et al., 2014), the MLP generator is an implicit generative model, whose predictive distribution is implicitly represented by its sampling process.

The energy Transformer is designed to accept continuous tokens as inputs. In the embedding layer, the lookup table is replaced with a linear projection, which maps each token of size $d_{\text{token}}$ to $d_{\text{model}}$. The representations extracted by Transformer are then mapped to $d_{mlp}$, serving as inputs for the MLP generator. The MLP generator also takes a random noise $\epsilon$ as an additional input to perturb the representation for the sampling purpose. The random noise of size $d_{\text{noise}}$ is drawn from a uniform distribution ranging from $[-0.5, 0.5]$, which is then embedded to size $d_{mlp}$.

The MLP generator consists of a few residual blocks that gradually inject noises into the prediction. Each residual block contains a two-layer FFN network with SiLU activation (Elfwing et al., 2018), and the noise is incorporated via

---

[2] Please note that the concept of "score" differs between scoring rules and score matching. In scoring rules, "score" is a measure used to assess the quality of probabilistic predictions. In score matching, "score" refers to the gradient $\nabla_x \log p(x)$.

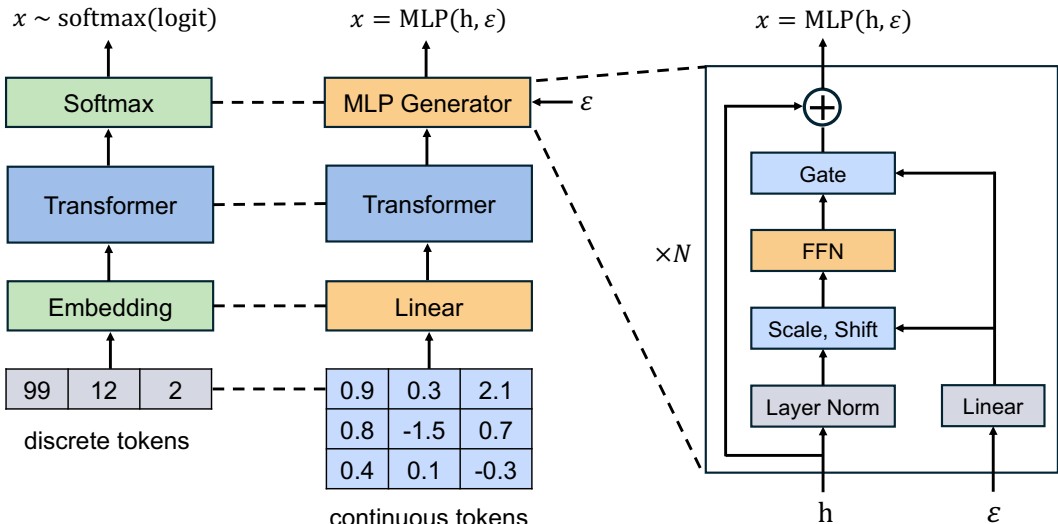

*Figure 1.* Comparison between the discrete-token standard Transformer and our continuous-token energy Transformer. At the input side, the embedding lookup table is replaced with a linear projection. At the output side, the softmax classification layer is replaced with a small MLP generator, which takes random noise $\epsilon$ as input to perturb the hidden state.

adaptive layer normalization (Peebles & Xie, 2023), which perturbs the prediction with shift, scale, and gate layers. Specifically, assuming that the input to the $i$-th residual block is $h^i$, its output is given by:

$$
\begin{aligned}
h^i_\epsilon &= (1 + scale(\epsilon)) \cdot LN(h^i) + shift(\epsilon), \\
h^{i+1} &= h^i + gate(\epsilon) \cdot FFN(h^i_\epsilon),
\end{aligned}
\tag{11}
$$

where $shift(\cdot)$, $scale(\cdot)$, and $gate(\cdot)$ are linear transformations that interpret the input noise as perturbation signals, $LN(\cdot)$ represents layer normalization, and $FFN(\cdot)$ denotes a two-layer feed-forward neural network with a intermediate dimension of $d_{mlp}$. Finally, the MLP generator concludes by predicting the next continuous token via a linear layer.

### 4.3. Other Techniques

In this section, we present several techniques that have proven effective in improving the visual generation quality of EAR.

**Temperature**. The temperature hyperparameter $\tau$ is widely used in the sampling process of generative models, which trades diversity for accuracy. For EAR, we can incorporate temperature hyperparameters during both training and inference, denoted as $\tau_{\text{train}}$ and $\tau_{\text{infer}}$, respectively. During training, the energy loss is composed of two components: $|x - y|^\alpha$ and $|x_1 - x_2|^\alpha$, where the latter measures the diversity of the generated outputs. Therefore, we can assign a weight $\tau_{\text{train}} < 1$ to $|x_1 - x_2|^\alpha$ in the energy loss, which can enhance the generation quality with a short period of fine-tuning. However, this approach is not applicable for

$\tau_{\text{train}} > 1$, as it would cause the loss function unbounded and hackable. During inference, directly modifying the scale of the noise would corrupt the generated images. Instead, we propose to only scale the $shift(\epsilon)$ by a temperature $\tau_{\text{infer}}$, while keeping $scale(\epsilon)$ and $gate(\epsilon)$ unchanged.

**Classifier-Free Guidance**. We employ Classifier-Free Guidance (CFG, Ho & Salimans, 2022) to improve the quality of conditional generation. At training time, we replace the condition with a dummy token for 10% of the samples. At inference time, the Transformer model is run with both the given condition and the dummy token, providing two outputs $h_c$ and $h_u$. The combination of the two outputs $h = \text{cfg} \cdot h_c + (1 - \text{cfg}) \cdot h_u$ is fed to the MLP generator, where cfg is the guidance scale. Following Chang et al. (2023), we linearly increase the guidance scale during the autoregressive generation. We sweep the optimal guidance scale for each model.

**Masked Autoregressive Generation**. Masked autoregressive models can be regarded as a type of autoregressive model that predicts a set of unknown tokens based on existing tokens (Chang et al., 2022; Li et al., 2023; Chang et al., 2023; Li et al., 2024). They supports bidirectional attention, which facilitates a more effective representation learning compared to causal attention. Consistent with Li et al. (2024), we find that masked autoregressive generation performs better than causal generation. During training, we randomly sample a masking ratio in the range of $[0.7, 1.0]$. During inference, we generate tokens in a random order, progressively reducing the masking ratio from 1.0 to 0 fol-

*Table 1.* Model comparisons on ImageNet 256×256 conditional generation. Metrics include Fréchet Inception Distance (FID), Inception Score (IS), Precision (Pre) and Recall (Rec). "↓" or "↑" indicate lower or higher values are better.

| Type | Model | #Params | w/o guidance | | | | w/ guidance | | | |
|------|-------|---------|------|------|------|------|------|------|------|------|
| | | | FID↓ | IS↑ | Pre↑ | Rec↑ | FID↓ | IS↑ | Pre↑ | Rec↑ |
| GAN | BigGAN (Brock et al., 2019) | 112M | 6.95 | 224.5 | 0.89 | 0.38 | - | - | - | - |
| | GigaGAN (Kang et al., 2023) | 569M | 3.45 | 225.5 | 0.84 | 0.61 | - | - | - | - |
| | StyleGan-XL (Sauer et al., 2022) | 166M | - | - | - | - | 2.30 | 265.1 | 0.78 | 0.53 |
| Diff | ADM (Dhariwal & Nichol, 2021) | 554M | 10.94 | 101.0 | 0.69 | 0.63 | 4.59 | 186.7 | 0.82 | 0.52 |
| | LDM-4† (Rombach et al., 2022) | 400M | 10.56 | 103.5 | 0.71 | 0.62 | 3.60 | 247.7 | 0.87 | 0.48 |
| | DiT-XL/2 (Peebles & Xie, 2023) | 675M | 9.62 | 121.5 | 0.67 | 0.67 | 2.27 | 278.2 | 0.83 | 0.57 |
| | L-DiT-7B (Alpha-VLLM, 2024) | 7B | 5.06 | 153.3 | 0.70 | 0.68 | 2.28 | 316.2 | 0.83 | 0.58 |
| | VDM++ (Kingma & Gao, 2023) | 2B | 2.40 | 225.3 | - | - | 2.12 | 267.7 | - | - |
| AR | VQGAN (Esser et al., 2021) | 1.4B | 15.78 | 78.3 | - | - | - | - | - | - |
| | RQ-Transformer (Lee et al., 2022) | 3.8B | 7.55 | 134.0 | - | - | - | - | - | - |
| | LlamaGen-3B (Sun et al., 2024a) | 3.1B | - | - | - | - | 2.18 | 263.3 | 0.84 | 0.54 |
| | MaskGIT (Chang et al., 2022) | 227M | 6.18 | 182.1 | 0.80 | 0.51 | - | - | - | - |
| | MAGE (Li et al., 2023) | 230M | 6.93 | 195.8 | - | - | - | - | - | - |
| | MAGVIT-v2 (Yu et al., 2024a) | 307M | 3.65 | 200.5 | - | - | 1.78 | 319.4 | - | - |
| | VAR-d30 (Tian et al., 2024) | 2.0B | - | - | - | - | 1.92 | 323.1 | 0.82 | 0.59 |
| | GIVT (Tschannen et al., 2023) | 304M | 5.67 | - | 0.75 | 0.59 | 3.35 | - | 0.84 | 0.53 |
| | MAR (Li et al., 2024) | 943M | 2.35 | 227.8 | 0.79 | 0.62 | 1.55 | 303.7 | 0.81 | 0.62 |
| EAR | EAR-B | 205M | 5.46 | 155.9 | 0.76 | 0.57 | 2.83 | 253.3 | 0.82 | 0.54 |
| | EAR-L | 474M | 3.69 | 183.4 | 0.77 | 0.59 | 2.37 | 273.8 | 0.81 | 0.57 |
| | EAR-H | 937M | 3.16 | 204.2 | 0.76 | 0.61 | 1.97 | 289.6 | 0.81 | 0.59 |

lowing a cosine schedule. By default, we use 64 generation steps in this schedule.

**Learning Rate for MLP Generator**. In our experiments using regular learning rates, we found that the model failed to converge. Given that the Transformer backbone remains unchanged, we hypothesize the MLP generator may need a smaller learning rate to ensure its training stability (Singh et al., 2015; Howard & Ruder, 2018; Xu et al., 2025). To address it, we adjust the MLP generator's learning rate by applying a constant multiplier $\lambda < 1$ throughout the training. Empirically, setting $\lambda = 0.25$ strikes a balance between training efficiency and stability.

## 5. Experiments

### 5.1. Settings

We evaluate visual generation capability on the class-conditional ImageNet 256×256 benchmark (Deng et al., 2009). We use Fréchet Inception Distance (FID, Heusel et al., 2017) as the main metric, and also provide Inception Score (IS, Salimans et al., 2016) and Precision/Recall (Kynkäänniemi et al., 2019) as secondary metrics. We follow the evaluation suite of Dhariwal & Nichol (2021).

We use the decoder-only Transformer architecture following the implementation in ViT (Dosovitskiy et al., 2021) for masked autoregressive generation. The class condition is represented as 64 class tokens at the start of the decoder sequence. We use the discrete VQ-16 tokenizer (Rombach et al., 2022) for the standrard Transformer and the continuous KL-16 tokenizer (Li et al., 2024) for our energy Transformer. The stride of both tokenizers is 16.

Following Li et al. (2024), we explore the scaling behavior of Energy-based AutoRegression (EAR) with three sizes of energy Transformer, referred to as EAR-B, EAR-L, and EAR-H, respectively. They respectively have 24, 32, 40 Transformer blocks and a width of 768, 1024, and 1280. The MLP generators account for approximately 15% of the parameter size of energy Transformer, which respectively have 6, 8, 12 blocks and a width of 1024, 1280, and 1536.

The random noise for the MLP generator has a size of $d_{noise} = 64$, independently drawn from a uniform distribution $[-0.5, 0.5]$ at each time step. We by default set $\alpha = 1$ to calculate the energy loss. We train our model for a total of 800 epochs, where the first 750 epochs use the standard energy loss and the last 50 epochs reduces the temperature $\tau_{train}$ to 0.99. The inference temperature $\tau_{infer}$ is set to

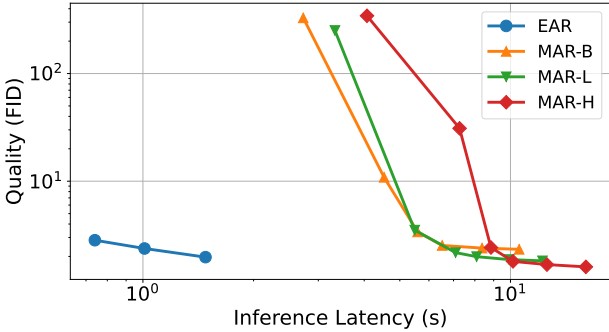

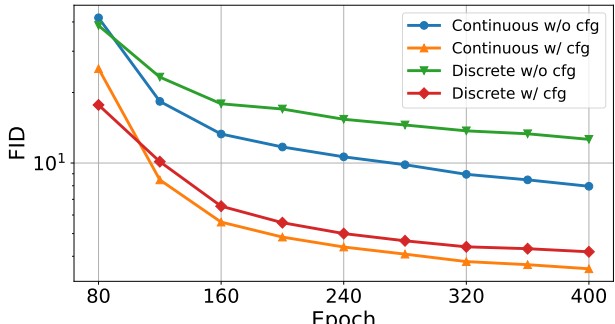

*Figure 2.* The speed/quality trade-off for EAR and MAR. The number of autoregressive steps is fixed at 64. For MAR, we vary the number of diffusion steps (10, 20, 25, 30, 40, 50) to generate outputs under different inference latencies. For EAR, the curve is obtained by using different model sizes (EAR-B, EAR-L, EAR-H). The inference time is measured on a single A100 GPU.

*Figure 3.* FID curves of the continuous-valued energy Transformer (205M) and the discrete-valued standard Transformer (196M). The guidance scale is 3.0.

0.7. Our models are optimized by the AdamW optimizer (Loshchilov & Hutter, 2019) with $\beta_1 = 0.9, \beta_2 = 0.95$. The batch size is 2048. The learning rate is 8e-4 and the constant learning rate schedule is applied with linear warmup of 100 epochs. We use a weight decay of 0.02, gradient clipping of 3.0, and dropout of 0.1 during training. Following Peebles & Xie (2023); Li et al. (2024), we maintain the exponential moving average of the model parameters with a momentum of 0.9999.

### 5.2. Main Results

In Table 1, we compare EAR with popular image generation models, including GANs, diffusion models, and VQ-based autoregressive models. Notably, EAR-B obtains a strong FID of 2.83 with only 205M parameters, and EAR-H achieves a competitive FID of 1.97, while maintaining a relatively modest model size among the leading systems. The scalability of EAR suggests that the generation quality could be further boosted through scaling.

Our approach is most closely aligned with MAR (Li et al., 2024), as they both model the distribution of continuous tokens with an MLP module on top of a masked autoregressive Transformer. They can both be viewed as instances under the Continuous VAR framework, with EAR and MAR maximizing the energy score and the Hyvärinen score, respectively. Figure 2 illustrates the inference latency (average time to generate an image) and generation quality (measured by FID) of the two methods. EAR is significantly more efficient in inference, capable of producing a high-quality image in roughly 1s, while MAR takes nearly 10 times longer to produce images of comparable quality. The efficiency advantage stems from the difference in probabilistic modeling. Trained with the diffusion loss, MAR necessitates multiple

denoising iterations to recover the target distribution. Conversely, the energy-style supervision enables EAR to make predictions within a single forward computation.

The continuous tokenizer we use exhibits a strong reconstruction quality of 1.22 FID. In contrast, a VQ tokenizer with the same model architecture only achieves a reconstruction FID of 5.87 (Rombach et al., 2022), which could become a bottleneck in generation quality. In Figure 3, we compare our continuous-valued energy Transformer with a discrete-valued Transformer that uses the VQ tokenizer. The results show that continuous tokenization with the energy loss consistently outperforms discrete tokenization with the cross-entropy loss, highlighting the great potential of Continuous VAR.

For causal autoregressive modeling, our findings align with Li et al. (2024) that both continuous- and discrete-valued Transformers can only achieve FID scores around 20. We hypothesize that causal models may suffer from overfitting due to the absence of random masking mechanism during training–a key feature of masked autoregressive modeling that enhances generalization.

### 5.3. Importance of Being Strictly Proper

In the energy loss presented in Equation 10, the exponential coefficient $\alpha$ was empirically set to 1 in previous experiments. While any choice of $\alpha \in (0, 2)$ ensures the strict propriety, energy losses with $\alpha < 1$ invariably induce rapid training collapse due to gradient instability. For example, the gradient of $|x_1 - x_2|^\alpha$ can be expressed as $\frac{\partial |x_1 - x_2|^\alpha}{\partial \theta} = \sum_{i=1}^n \frac{\alpha(x_1^i - x_2^i)}{|x_1 - x_2|^{2-\alpha}} \cdot \frac{\partial (x_1^i - x_2^i)}{\partial \theta}$. When training begins, independent samples $x_1$ and $x_2$ are typically nearly indistinguishable, which causes the denominator $|x_1 - x_2|^{2-\alpha}$ to approach zero exponentially faster than the numerator when $\alpha < 1$, resulting in unbounded gradient magnitudes that destabilize optimization.

Additionally, while the energy score remains proper at $\alpha = 2$, its non-strict propriety (only expectation alignment $\mathbb{E}_p[x] = \mathbb{E}_q[y]$ is enforced) proves insufficient for effective training. As evidenced in Table 2, training with $\alpha = 2$ fails to generate meaningful contents (FID $> 100$), whereas leveraging strictly proper energy scores with $\alpha \in [1, 2)$ achieves decent generation quality. It validates our claim on the importance of being strictly proper for scoring rules, which guarantees a unique global minimum.

Table 2. The performance of EAR-B when being trained with different exponential term $\alpha$ in the energy loss. The number of training epochs is 400. The guidance scale is 3.0.

| $\alpha$ | 1.0 | 1.25 | 1.5 | 1.75 | 2.0 |
|---|---|---|---|---|---|
| FID | 3.55 | 3.73 | 4.10 | 4.32 | 188.1 |
| IS | 230.3 | 223.1 | 212.1 | 204.2 | 6.4 |

### 5.4. Importance of Being Expressive

While the logarithmic score also serves as a strictly proper scoring rule, it necessitates explicit knowledge of the predictive probability density, which poses a challenge for continuous-valued models. To obtain an explicit likelihood estimation, it generally requires constraining the predictive distribution of the model, as exemplified in Tschannen et al. (2023). For instance, consider a model that parameterizes a Gaussian distribution with mean vector $\mu$ and covariance matrix $\Sigma$. The corresponding negative log-likelihood objective becomes:

$$\mathcal{L} = -\log f(x|\mu, \Sigma) = \frac{1}{2}(x - \mu)^T \Sigma^{-1} (x - \mu) \\ + \frac{n}{2}\log(2\pi) + \frac{1}{2}\log|\Sigma|. \tag{12}$$

This objective reduces to the Mean Squared Error (MSE) loss under the common assumption of channel independence with a fixed standard deviation $\sigma$. We employ this MSE loss to train a Gaussian Transformer and experiment with different $\sigma$ during inference. The results are depicted in Figure 4.

As seen, an appropriate variance selection yields non-trivial generation quality, but the performance gap compared to EAR remains substantial, suggesting that the token distribution is complex and challenging to be explicitly represented using predefined distributions. Our proposed energy Transformer addresses this challenge through its inherently expressive architecture: the model implicitly defines the predictive distribution through its sampling process, which enables the automatic learning of complex data distributions without restrictive prior assumptions.

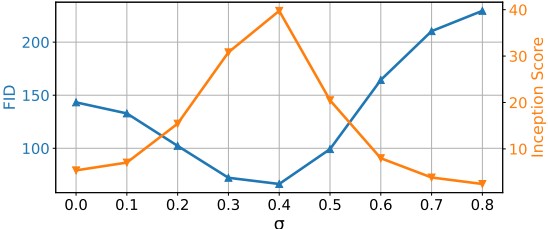

Figure 4. Generation quality of the Gaussian Transformer under different standard deviations during inference. The model size is 184M. cfg is disabled since it does not work well here.

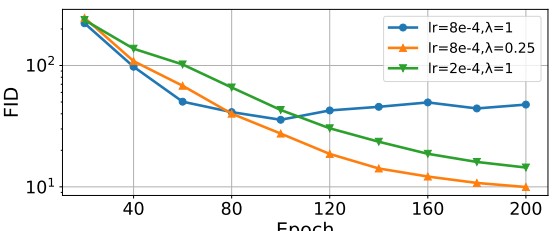

Figure 5. The results of varying learning rates for EAR-B.

### 5.5. Ablation Study

**Effect of Learning Rate.** We observed that the model failed to converge when using standard learning rates, and reducing the learning rate specifically for the MLP generator was found effective to enhance training stability. As illustrated in Figure 5, when the entire model was trained with a global learning rate $lr = 8e - 4$, the training process eventually collapsed after several epochs. By selectively lowering the learning rate for the MLP generator to $\lambda = 0.25\times$, the training process is stabilized. While reducing the learning rate of the entire model also enables successful training, this approach results in relatively lower training efficiency.

**Effect of Noise.** In our experiments, we observed that both the type and dimension of random noise can affect model performance. We experimented with Uniform noise and Gaussian noise, setting the noise dimension, $d_{noise}$, to 32, 64, and 128. As shown in Table 3, uniform noise consistently outperforms Gaussian noise, and $d_{noise} = 64$ performs better than other settings. Therefore, we adopt the 64-dimensional uniform noise in EAR.

Table 3. The results of varying random noises for EAR-B. The number of training epochs is 400. The guidance scale is 3.0.

| $d_{noise}$ | Uniform | | | Gaussian | | |
|---|---|---|---|---|---|---|
| | 32 | 64 | 128 | 32 | 64 | 128 |
| w/o cfg | 9.87 | 7.95 | 7.04 | 9.45 | 7.79 | 7.17 |
| w/ cfg | 3.89 | 3.55 | 4.34 | 4.01 | 3.60 | 4.51 |

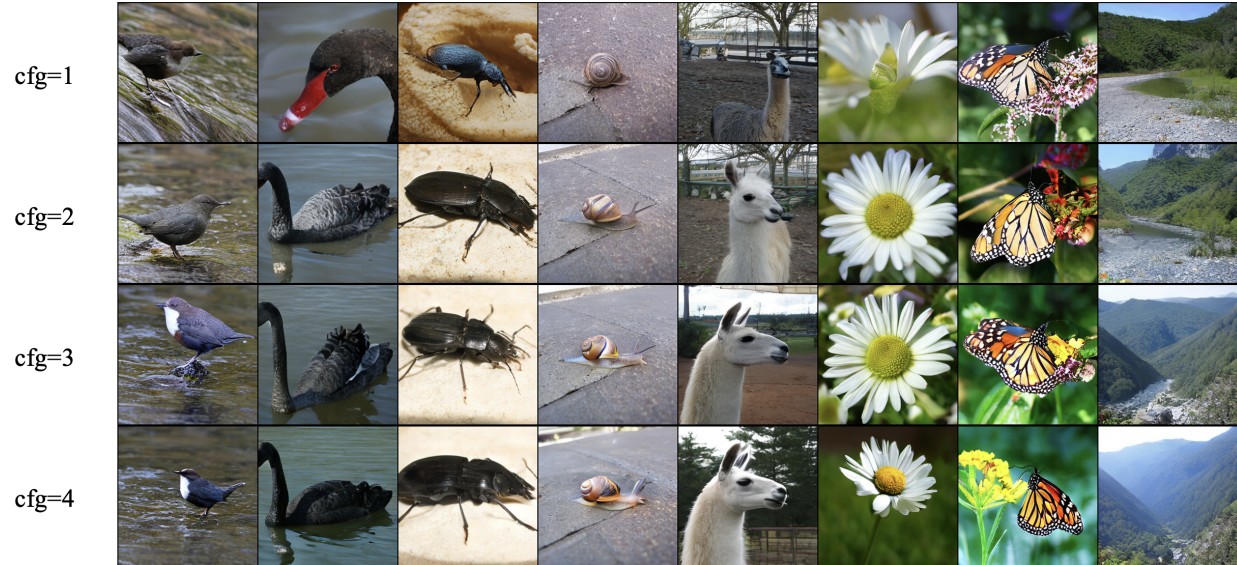

Figure 6. Samples of EAR-H under different gudiance scales. We fix the random seed and apply the constant cfg schedule during sampling.

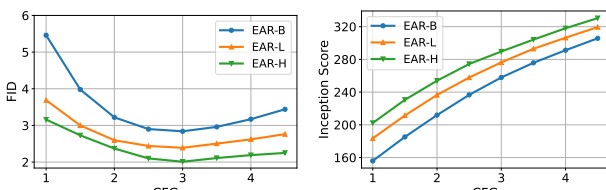

Figure 7. The results of varying classifier-free guidance scales for EAR models.

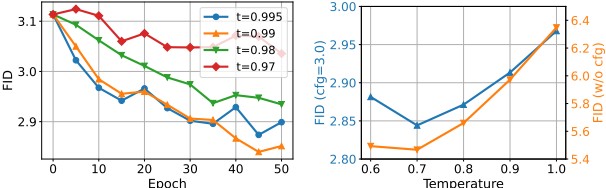

Figure 8. The results of varying temperatures $\tau_{\text{train}}$ (left) and $\tau_{\text{infer}}$ (right) for EAR-B.

**Effect of CFG.** Classifier-free guidance (Ho & Salimans, 2022) plays a crucial role in the inference stage of EAR. Figure 7 illustrates the variations of FID and Inception Score across different cfg scales. The image quality, as measured by the Inception Score, consistently improves with increasing cfg. However, the FID metric reaches its optimal value around cfg=3.0, as a excessive guidance scale can compromise the generation diversity. Figure 6 illustrates the sampling outputs of EAR-H under different guidance scales, where a larger cfg scale generally produces more fine-grained images.

**Effect of Temperature.** We employ temperature hyperparameters $\tau_{\text{train}}$ and $\tau_{\text{infer}}$ to trade diversity for accuracy. Figure 8 shows the impact of varying temperatures during the fine-tuning and inference of EAR-B. These results induce a temperature combination of $\tau_{\text{train}} = 0.99$ and $\tau_{\text{infer}} = 0.7$.

## 6. Conclusion

This paper introduces a Continuous VAR framework that enables direct visual autoregressive generation without vector quantization. The Continuous VAR framework is grounded in strictly proper scoring rules, and we primarily explore a class of energy-based training objectives, which is likelihood-free and induces an expressive enery Transformer architecture. Our experimental results demonstrate competitive performance in both generation quality and inference efficiency, while leaving substantial room for future improvements. Promising research directions include: 1) architectural optimization of the energy Transformer, 2) incorporation of alternative strictly proper scoring rules as training objectives, 3) extension to more continuous modalities such as video and audio, and 4) continuous language modeling through the conversion of discrete text into latent vector representations.

## Impact Statement

This paper presents work whose goal is to advance the field of Machine Learning. There are many potential societal consequences of our work, none which we feel must be specifically highlighted here.

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

# A. Additional Results

Table 4. Model comparisons on ImageNet 512×512 conditional generation. The cfg scale is set to 4.0.

| Type | Model | #Params | w/o guidance | | w/ guidance | |
|---|---|---|---|---|---|---|
| | | | FID↓ | IS↑ | FID↓ | IS↑ |
| Diff | ADM (Dhariwal & Nichol, 2021) | 554M | 23.24 | 58.1 | 7.72 | 172.7 |
| | DiT-XL/2 (Peebles & Xie, 2023) | 675M | 12.03 | 105.3 | 3.04 | 240.8 |
| | VDM++ (Kingma & Gao, 2023) | 2B | 2.99 | 232.2 | 2.65 | 278.1 |
| AR | MaskGIT (Chang et al., 2022) | 227M | 7.32 | 156.0 | - | - |
| | MAGVIT-v2 (Yu et al., 2024a) | 307M | 3.07 | 213.1 | 1.91 | 324.3 |
| | GIVT (Tschannen et al., 2023) | 304M | 8.35 | - | - | - |
| | MAR (Li et al., 2024) | 481M | 2.74 | 205.2 | 1.73 | 279.9 |
| EAR | EAR-B | 205M | 7.75 | 141.5 | 3.38 | 227.0 |

Table 5. The effect of attention masking on EAR-B. The number of training epochs is 400.

| Type | w/o guidance | | w/ guidance | |
|---|---|---|---|---|
| | FID↓ | IS↑ | FID↓ | IS↑ |
| Causal | 17.83 | 78.6 | 8.10 | 144.5 |
| Bidirection | 7.95 | 130.5 | 3.55 | 230.3 |

