# OpenReview forum: "Continuous Visual Autoregressive Generation via Score Maximization"
_ICML.cc/2025/Conference — ICML 2025 poster_

### Official Review · Reviewer_npsM · 2025-03-11

**Overall Recommendation:** 4

**Summary:**

This paper presents a new continuous visual autoregressive generative framework, which prevents the information loss caused by vector quantization. This framework takes energy score as the training objectives, which is likelihood-free and easy to make probabilistic predictions in the continuous space. In addition, the authors conduct many experiments to demonstrate the effectiveness of the proposed method.

## update after rebuttal
I appreciate the authors' thorough response and the effort they put into the rebuttal. My concerns have been resolved, thus, I raise my score to accept. Last but not least, the authors are encouraged to include experiments on ​ImageNet $512 \times 512$ conditional generation in the final version.

**Claims And Evidence:**

Strengths
+ The motivation of this paper is strong and meaningful. It is an imperative direction for visual generation.
+ Unlike quantization-based approaches, the proposed method can avoid the information loss. It is mainly supported by the superior results in Table 1.
+ Based on energy score, autoregressive models can achieve an efficient inference, due to it only needs a single forward pass for each token prediction.

Weaknesses

None

**Essential References Not Discussed:**

To my knowledge, there is no other references to be discussed.

**Experimental Designs Or Analyses:**

Strengths
+ The conducted experiments can demonstrate the effectiveness of the proposed method. In addition, the ablation studies on $\alpha$ and CFG also highlight the suitable value of these hyperparameters.

Weaknesses
- In Table 1, it is essential to show the results of MAR and GIVT. They are two important baselines to compare.
- In Figure 2, the speed for MAR maybe not correct. To my knowledge, the speed of MAR in Figure 2 is slower than MAR paper. Could you provide more details about inference settings?
- This paper only shows the comparisons on ImageNet $256 \times 256$ conditional generation. It is better to show more results to assess the proposed method, like ImageNet $512 \times 512$ conditional generation.
- The ablation studies are insufficient. It is better to investigate more settings, such as the choice of random noise and mask ratio.

**Methods And Evaluation Criteria:**

Strengths
+ The proposed energy loss is simple and intuitive. It addresses the shortcomings of GIVT and diffusion loss.
+ The proposed method is easy to implement and easy to understand. Its effectiveness is validated by the results on ImageNet $256 \times 256$ conditional generation.

Weaknesses
- The random noise is drawn from a uniform distribution. What the principle for this choice is? If using a gaussian noise, is it ok?
- Likewise, the masking ratio is sampled in the range of $[0.7, 1.0]$. It is not a default choice. What the principle for this choice is?

**Other Comments Or Suggestions:**

None

**Other Strengths And Weaknesses:**

None

**Questions For Authors:**

I appreciate the effectiveness of the proposed method on image generation, although the experiments of this paper are insufficient. Thus, I prefer to rate this paper as weak accept. If authors addresses my concerns, I would like to raise my score.

**Relation To Broader Scientific Literature:**

To my knowledge, the proposed method in this paper is new.

**Theoretical Claims:**

I have check the correctness of any proofs for theoretical claims. And I do not find any errors.

---

> ### Author Rebuttal · Authors · 2025-03-27
>
> We sincerely appreciate the Reviewer's time and efforts in reviewing our work. We provide discussions about the concerns as follows.
>
> > In Table 1, it is essential to show the results of MAR and GIVT. They are two important baselines to compare.
>
> We fully agree with the Reviewer and will include comparisons with ​MAR and ​GIVT in the main table.
>
> > In Figure 2, the speed for MAR maybe not correct. To my knowledge, the speed of MAR in Figure 2 is slower than MAR paper. Could you provide more details about inference settings?
>
> The difference in reported speeds arises from differing evaluation protocols:
>
> + In the MAR paper, the authors measure inference time using the ​maximum batch size and report the ​average time per image, which benefits from parallel computation.
>
> + In our experiments, we follow the more common practice of benchmarking with a ​batch size of 1 to reflect real-world deployment scenarios where latency matters.
>
> Under the setting of batch=1, MAR’s diffusion process incurs significant decoding latency due to its iterative nature. While increasing the batch size improves MAR’s throughput, EAR remains approximately ​2× faster than MAR under maximum batch size conditions.
>
> > This paper only shows the comparisons on ImageNet 256x256 conditional generation. It is better to show more results to assess the proposed method, like ImageNet  conditional generation.
>
> We appreciate the Reviewer’s valuable suggestion. We are currently conducting experiments on ​ImageNet 512×512 conditional generation to further validate the effectiveness of our method. These results will be included in the final version of our paper.
>
> > The ablation studies are insufficient. It is better to investigate more settings, such as the choice of random noise and mask ratio.
>
> We thank the Reviewer for pointing out these concerns. Regarding the mask ratio, we directly adopted the range of [0.7, 1] from MAR, without further tuning. Regarding the random noise, we empirically observe that uniform noise yields slightly better performance compared to Gaussian noise. Specifically, for EAR-B: w/o CFG, FID increases from ​5.46 (uniform) to ​5.74 (Gaussian); w/ CFG, FID increases from ​2.83 (uniform) to ​2.89 (Gaussian). We will ablate this choice in the revised manuscript.

---

### Official Review · Reviewer_1FmY · 2025-03-13

**Overall Recommendation:** 2

**Summary:**

This paper introduces a continuous visual autoregressive framework EAR. The approach is grounded in strictly proper scoring rules, which provide a statistical basis for evaluating generative models, and primarily utilizes an energy score-based training objective to handle continuous data without requiring likelihood estimation.

Update after rebuttal: I appreciate the authors' efforts. The response has clarified some of the issues I raised and helped me better understand certain aspects of the work. However, taking into account the points raised by other reviewers, I do not find enough justification to change my overall assessment. Therefore, I will maintain my original rating.

**Claims And Evidence:**

Not clear enough. See weaknesses.

**Essential References Not Discussed:**

No.

**Experimental Designs Or Analyses:**

Yes.

**Methods And Evaluation Criteria:**

Yes.

**Other Comments Or Suggestions:**

See weaknesses.

**Other Strengths And Weaknesses:**

Strengths:
1. The paper is the first to apply strictly proper scoring rules to continuous visual autoregressive modeling.
2. It offers a theoretical discussion on how different strictly proper scores, such as the energy score, can be leveraged as training objectives, unifying previous continuous autoregressive approaches like GIVT and diffusion loss under a common framework.

Weaknesses:
1. The paper does not clearly justify why Strictly Proper Scoring Rules are used to analyze Continuous Visual Autoregressive Generation. While the paper mentions that “GIVT is confined to the pre-defined family of Gaussian mixtures,” this is not necessarily a drawback. Instead, it may be the result of deliberate selection by the GIVT authors after thorough comparisons. The paper fails to explain why this limitation is a concern and to motivate its ideas.
2. The authors claim that continuous AR is superior to discrete AR, but the performance of their proposed EAR (Table 1) does not consistently support this assertion. For instance, EAR underperforms compared to MAGVIT-v2. Moreover, the training cost of EAR is significantly higher than existing approaches: EAR requires 800 training epochs, whereas VAR only needs 200 to 350 epochs.
3. The experiments in pages 7-8 mainly include obvious experiments or hyperparameter tuning studies, without exploring more critical aspects, for example, temperature or random noise in the MLP generator.

**Questions For Authors:**

See weaknesses.

**Relation To Broader Scientific Literature:**

The main framework is from VAR. This paper modifies some components.

**Theoretical Claims:**

Yes. Eq (1) - Eq (7) are from existing papers.

---

> ### Author Rebuttal · Authors · 2025-03-27
>
> We sincerely appreciate the Reviewer's time and efforts in reviewing our work. We provide discussions about the concerns as follows.
>
> > While the paper mentions that “GIVT is confined to the pre-defined family of Gaussian mixtures,” this is not necessarily a drawback. Instead, it may be the result of deliberate selection by the GIVT authors after thorough comparisons. The paper fails to explain why this limitation is a concern and to motivate its ideas.
>
> We appreciate the reviewer’s thoughtful feedback. While we acknowledge that the Gaussian mixture assumption in GIVT may indeed represent a deliberate and well-justified choice for modeling continuous tokens, we respectfully disagree with the assertion that this assumption "is not necessarily a drawback".
>
> The fundamental limitation arises when using likelihood maximization to train continuous autoregressive models: the predictive distribution must be constrained to a predefined parametric family (e.g., Gaussian mixtures) to enable tractable likelihood estimation. While Gaussian mixtures may offer the best fit within this constrained framework, this approach inherently restricts the model’s ability to represent arbitrary target distributions—a limitation that cannot be fully mitigated by the choice of distribution family.
>
> In contrast, our proposed EAR framework circumvents this challenge through its inherently expressive architecture. By implicitly defining the predictive distribution via its sampling process, EAR avoids restrictive parametric assumptions and enables automatic learning of complex data distributions. We will clarify this distinction more explicitly in the revised manuscript to better motivate our method.
>
> > The authors claim that continuous AR is superior to discrete AR, but the performance of their proposed EAR (Table 1) does not consistently support this assertion. For instance, EAR underperforms compared to MAGVIT-v2. Moreover, the training cost of EAR is significantly higher than existing approaches: EAR requires 800 training epochs, whereas VAR only needs 200 to 350 epochs.
>
> We appreciate the reviewer’s attention to the empirical results. Our claim regarding the advantages of continuous AR modeling is primarily supported by the comparison in ​Figure 3, where EAR outperforms its discrete counterpart (using the VQ tokenizer in [1]). This demonstrates the benefits of continuous modeling when controlling for other architectural factors.
>
> Regarding MAGVIT-v2, we acknowledge that EAR currently uses a ​three-year-old continuous tokenizer architecture, which may limit its performance compared to state-of-the-art discrete tokenizers. We hypothesize that integrating a ​more advanced, fine-grained continuous tokenizer could better showcase the potential of continuous AR modeling.
>
> As for training efficiency, we agree that EAR currently requires more epochs due to the ​inherent complexity of learning continuous distributions without restrictive parametric assumptions. Future work could explore optimizations to reduce training costs while retaining EAR’s expressiveness.
>
> [1] High-Resolution Image Synthesis with Latent Diffusion Models.
>
> > The experiments in pages 7-8 mainly include obvious experiments or hyperparameter tuning studies, without exploring more critical aspects, for example, temperature or random noise in the MLP generator.
>
> We thank the Reviewer for pointing out these concerns. Regarding the temperature, we have tuned it on EAR-B and found that τ=0.98 works better than other choices. Regarding the random noise, we have conducted experiments comparing uniform noise and Gaussian noise, and observed that uniform noise leads to marginally better performance. We will include these ablation experiments in the revised manuscript.

---

### Official Review · Reviewer_gkkh · 2025-03-17

**Overall Recommendation:** 3

**Summary:**

This paper introduce energy-based autoregressive to train a continuous autoregressive models. The continuous model bypasses the traditional approach of using discrete representation to train an autoregressive model, therefore reduce the information loss during discrete quantization. The experiment shows the effectiveness of continuous autoregressive framework on imagenet compared to other models.

## update after rebuttal
The novelty has partially resolved my concern. Therefore, I increase my score to weak accept. Please include the ablation for causal and full attention in your revised paper.

**Claims And Evidence:**

The paper is well-written and provide evidence to most of their claims.

**Essential References Not Discussed:**

The paper mentions all essential references.

**Experimental Designs Or Analyses:**

The experimental designs are sound and valid.

**Methods And Evaluation Criteria:**

The evaluation follows standard protocol

**Other Comments Or Suggestions:**

No

**Other Strengths And Weaknesses:**

**Strength**

1. The idea of using strictly proper score as loss function for continuous autoregressive seems novel to me. This is the first work investigating this idea in visual autoregressive model.

2. The paper writing is clear and easy to understand.

**Weakness**:

1. The ablation for masked autoregressive should be provided with causal and full attention like setting in MAR [1].

2. The experiment in table 1 lacks of comparison with continuous AR like [1], [2]. The paper states that in [2] GMM could limit expressivity line 155 2nd column. However, the performance of proposed method still lag behind GIVT. The author should include the performance of [1] and [2] in the main table and make a comprehensive comparison.

3. Why the random noise in MLP is uniform but not normal. Why uniform is better choice ?. The author should ablate this choice.

4. The inference process is significantly affected by variance $\sigma$ in section 5.4. This could be similar to Truncated trick in Big GAN. How the author think about it ?. I would like to hear the author opinion.

[1]: Autoregressive Image Generation without Vector Quantization

[2]: GIVT: Generative Infinite-Vocabulary Transformers

**Questions For Authors:**

Please refer to the weakness

**Relation To Broader Scientific Literature:**

The strictly proper scoring rule is interesting, the paper is one of first work exploring this as loss for training autoregressive model.

**Theoretical Claims:**

There is no theoretical proof. The paper introduces strictly proper scoring rules and directly apply it to train autoregressive models.

---

> ### Author Rebuttal · Authors · 2025-03-27
>
> We sincerely appreciate the Reviewer's time and efforts in reviewing our work. We provide discussions about the concerns as follows.
>
> > The ablation for masked autoregressive should be provided with causal and full attention like setting in MAR.
>
> We thank the Reviewer for raising this concern. We will enrich the ablation studies and provide comparisons between causal and full attention.
>
> > The experiment in table 1 lacks of comparison with continuous AR like [1], [2]. The paper states that in [2] GMM could limit expressivity line 155 2nd column. However, the performance of proposed method still lag behind GIVT. The author should include the performance of [1] and [2] in the main table and make a comprehensive comparison.
>
> We will follow the suggestion and include comparisons with continuous AR models (MAR and GIVT) in the main table. We would also like to respectfully clarify that our method (EAR) does not lag behind GIVT in performance. Specifically:
>
> + Base-scale models: GIVT (304M params) achieves an FID of ​3.35, while our EAR-B (205M params) achieves ​2.83.
>
> + Large-scale models: GIVT-L (1.67B params) achieves an FID of ​2.59, while our EAR-H (937M params) achieves ​1.97.
>
> > Why the random noise in MLP is uniform but not normal. Why uniform is better choice ?. The author should ablate this choice.
>
> We empirically observe that uniform noise yields slightly better performance compared to Gaussian noise. Specifically, for EAR-B: w/o CFG, FID increases from ​5.46 (uniform) to ​5.74 (Gaussian); w/ CFG, FID increases from ​2.83 (uniform) to ​2.89 (Gaussian). We will report this comparison in the revised manuscript.
>
> 4. The inference process is significantly affected by variance σ in section 5.4. This could be similar to Truncated trick in Big GAN. How the author think about it?
>
> We would like to clarify that Section 5.4 actually presents results using an MSE loss function (rather than our proposed EAR method), which can be interpreted as modeling continuous token distributions with Gaussian distributions. In this setup, the standard deviation σ is specified during inference.
>
> Our experiments show optimal performance occurs around σ=0.4, likely because this value best matches the inherent variance of the continuous token distribution. The reviewer's analogy to BigGAN's Truncation Trick is indeed insightful - they share the commonality of potentially trading sample diversity for sample quality. The Truncation trick operates on latent space sampling, while the variance adjustment operates more directly on the predicted distribution's shape.

---

### Decision · Program_Chairs · 2025-05-01

**Decision:**

Accept (poster)

**Comment:**

The paper introduces a novel framework, Energy-based Autoregression (EAR), for continuous visual autoregressive generation. It uses strictly proper scoring rules to enable direct, quantization-free autoregressive modeling of continuous visual data. By formulating training objectives based on the energy score, the proposed method achieves superior performance and avoids the limitations associated with existing continuous autoregressive models using Gaussian mixtures or diffusion models to model the per-token continuous distribution. Reviewers highlighted the strong theoretical grounding, methodological novelty, and compelling empirical results on ImageNet. Despite minor initial concerns regarding ablations and comparative experiments, the authors effectively addressed these points in their rebuttal. Given the above reasons, I recommend acceptance of this paper.